# The Modulation of Cognitive Performance with Transcranial Alternating Current Stimulation: A Systematic Review of Frequency-Specific Effects

**DOI:** 10.3390/brainsci10120932

**Published:** 2020-12-02

**Authors:** Katharina Klink, Sven Paßmann, Florian H. Kasten, Jessica Peter

**Affiliations:** 1Graduate School for Health Sciences, University of Bern, 3000 Bern, Switzerland; katharina.klink@upd.unibe.ch; 2University Hospital of Old Age Psychiatry and Psychotherapy, University of Bern, 3000 Bern, Switzerland; jessica.peter@upd.unibe.ch; 3Division of Cognitive Biopsychology and Methods, Department of Psychology, University of Fribourg, 1700 Fribourg, Switzerland; 4Experimental Psychology Lab., Department of Psychology, European Medical School, Cluster for Excellence “Hearing for All”, Carl von Ossietzky University, 26129 Oldenburg, Germany; florian.kasten@uni-oldenburg.de

**Keywords:** transcranial alternating current stimulation, tACS, cognitive performance, systematic review

## Abstract

Transcranial alternating current stimulation (tACS) is a non-invasive brain stimulation technique that allows the manipulation of intrinsic brain oscillations. Numerous studies have applied tACS in the laboratory to enhance cognitive performance. With this systematic review, we aim to provide an overview of frequency-specific tACS effects on a range of cognitive functions in healthy adults. This may help to transfer stimulation protocols to real-world applications. We conducted a systematic literature search on PubMed and Cochrane databases and considered tACS studies in healthy adults (age > 18 years) that focused on cognitive performance. The search yielded *n* = 109 studies, of which *n* = 57 met the inclusion criteria. The results indicate that theta-tACS was beneficial for several cognitive functions, including working memory, executive functions, and declarative memory. Gamma-tACS enhanced performance in both auditory and visual perception but it did not change performance in tasks of executive functions. For attention, the results were less consistent but point to an improvement in performance with alpha- or gamma-tACS. We discuss these findings and point to important considerations that would precede a transfer to real-world applications.

## 1. Introduction

Transcranial alternating current stimulation (tACS) is increasingly being used as a method to modulate cognitive performance [1]. tACS non-invasively applies a sinusoidal oscillating current to modulate intrinsic oscillatory activity [2]. This temporally aligns neural firing and, consequently, entrains endogenous oscillations in the brain [3,4,5]. In addition, tACS may induce longer lasting synaptic changes through spike-time-dependent plasticity [6,7].

The modulation of intrinsic oscillations with tACS can be done in several ways [2], as shown in Figure 1: (A) Applying tACS at the same frequency as the intrinsic oscillation will lead to an increase in amplitude (Figure 1A); (B) Stimulating with a certain frequency may also affect the oscillations in other frequency bands (e.g., with so-called cross-frequency coupling; Figure 1B); (C) Stimulation electrodes that deliver alternating current synchronized (in-phase, Figure 1C) or de-synchronized (anti-phase, Figure 1C) induce or disturb the phase coherence between uni- and bihemispheric cortical activity; (D) Applying tACS slightly above or below the intrinsic frequency induces an acceleration or deceleration of the intrinsic oscillation (Figure 1D). This process of entrainment—that is, the synchronization of intrinsic oscillations to an externally applied frequency—is suggested to explain tACS’ effects on cognitive performance when applied during a task.

For the modulation of cognitive performance with tACS, online effects can be distinguished from offline effects. By definition, entrainment occurs during stimulation (i.e., online)—since the internal oscillation is coupled with an external oscillator. However, successful entrainment likely induces longer lasting aftereffects (i.e., offline effects; Figure 1E). These aftereffects are suggested to reflect plasticity-related network changes rather than entrainment per se [8]. To test online effects on cognitive performance, the timing of stimuli presentation can be adapted to fit to postulated prerequisites of oscillations. Thus, tACS-induced behavioral changes are measured directly. On the other hand, applying tACS prior to or between tasks aims at measuring the offline effects of stimulation on cognitive performance (Figure 1E).

Some cognitive processes have been, more or less specifically, related to certain frequency bands [9]. Delta (0–4 Hz) has traditionally been associated with deep sleep and memory consolidation [10] and, only during recent years, it has been associated with attention [11] or with cognitive processes that rely on rhythmicity, e.g., certain motor functions [10]. Theta (4–7 Hz) oscillations have been mainly associated with working memory [12,13] and episodic memory [14]. Alpha (8–13 Hz) has been linked to executive functions [15], visual attention [16], and to memory processes [17]. Beta (13–30 Hz) has been related to motor functions and attention [18] as well as to working memory and executive control [19]. Gamma (30–80 Hz) has been linked to the processing of incoming information [20] as well as to working memory [21] and episodic memory [22]. Additionally, it plays a role in auditory perception [23].

Although several literature overviews of tACS studies are available, they only rarely address specific effects on cognitive performance, but see [24] for a meta-analysis. Instead, they cover safety guidelines, e.g., [25], the underlying mechanisms, e.g., [26], technical issues [27], or reliability and validity [8,25]. Only two reviews, so far, attempted to provide an overview of outcomes for different cognitive domains [1,28] but they did not mention differences between on-/offline tACS, current intensities, or electrode montage. For a transfer of certain stimulation protocols to real-world applications, however, it would be important to identify parameters and settings showing reliable effects in the laboratory. Since the number of tACS studies is steadily increasing, we aim to give an updated, comprehensive overview of the efficacy of the different stimulation protocols used so far to manipulate behavioral changes in various cognitive domains. This may help to translate the most effective tACS protocols for each cognitive domain to real-world applications.

## 2. Materials and Methods

For the present systematic review, we followed the Preferred Reporting Items for Systematic Reviews and Meta-Analysis (PRISMA) guidelines [29]. We pre-registered this study with PROSPERO (CRD42020193621, https://www.crd.york.ac.uk/prospero).

We systematically searched for published studies in English with no date restriction in the following databases: PubMed (https://pubmed.ncbi.nlm.nih.gov/) and Cochrane Library (https://www.cochranelibrary.com/advanced-search). We used the following search terms:

“Transcranial AND alternating AND current AND stimulation AND *cognitive domain* AND healthy AND adults”“TACS AND *cognitive domain* AND healthy AND adults”.

We investigated the following cognitive functions: attention, intelligence, auditory perception, visual perception, perception, decision-making, procedural memory (i.e., motor learning), executive functions, long-term memory, working memory, episodic memory.

To be eligible for inclusion, studies needed to: (1) apply frequency-specific tACS, (2) focus on (at least) one of the above mentioned cognitive domains, (3) include healthy adults (>18 years or older) with a minimum of *n* = 15 for a within-subject or *n* = 20 for a between-subject design, and (4) stimulate continuously for at least 5 min.

Following the PRISMA guidelines, two authors (SP and KK) screened abstracts and titles independently and analyzed studies that met inclusion criteria. We additionally screened the reference lists of included studies to identify additional studies. A third author (JP) monitored the inclusion process at any stage (i.e., identification, screening, and eligibility). All authors made a final decision on studies included in the descriptive synthesis.

## 3. Results

The search yielded *n* = 109 studies, of which *n* = 57 met the inclusion criteria. Figure 2 shows the selection process (identification, screening, eligibility, and inclusion) with the respective number of studies [29].

A comprehensive overview of all study designs, stimulation parameters, outcomes as well as risk of bias for all included studies can be found in Table 1. The risk of bias was rated as low when the following criteria were met: the sample size was sufficiently high, a control condition was defined, and a detailed description of statistical analysis was given. Otherwise, the risk of bias was rated as high. Figure 3 shows the mean age of the participants (A left) as well as the sample sizes (B right) of all included studies.

In the following, we will more thoroughly describe the included studies for each cognitive domain.

### 3.1. Auditory Perception

Auditory perception is the ability to detect, localize, and identify sounds. Therefore, typical auditory perception tasks test the detection of auditory signals within background noise or the differentiation of at least two acoustic signals.

In Riecke and colleagues [50], delta/theta-tACS (4 Hz) was applied during a stream perception task in which the participants continuously needed to search for streams in ongoing background noise. The authors varied the strength of the phase synchrony between the target stream in the ear and the electric stimulation. They found that streams emerged more rapidly from a noisy background when they were more synchronous with endogenous delta/theta neuronal oscillations entrained by tACS. In a follow-up study [51], the same authors again examined delta/theta-tACS (4 Hz) during another speech perception task. This study differed from the first in that participants had to recognize words in the speech of one out of two simultaneous speakers (session one) or in one speaker only (session two). For session one, the authors found an increase in accuracy when tACS and speech rhythm were aligned. In session two, the speech rhythm was intentionally removed from the one speaker signal. Increased accuracy was found only when tACS phase preceded speech, instead of being directly aligned with the auditory signal.

Zoefel and colleagues [48,49] applied a speech perception task with rhythmically spoken five-syllable sentences in two conditions: in one condition, these sentences were clearly identifiable as speech, while in the other condition, they sounded like noise that, although it resembled speech, could not be recognized in isolation. In the first experiment [48], they applied delta/theta-tACS (3.125 Hz) while the participants were asked to detect small shifts in stimulus rhythm. Delta/theta-tACS (3.125 Hz) had no effect on detection performance. In a follow-up study with the same stimuli [49], the participants had to detect words with an alternative forced choice task. In this study, the authors compared unilateral to bilateral stimulation, applying delta/theta-tACS (3.125 Hz) during the speech perception task. Compared with sham, the bilateral stimulation improved word detection only when speech and tACS were aligned, while performance was disrupted when speech and tACS were not aligned. Unilateral delta/theta-tACS (3.125 Hz) had no effects on word detection.

Rufener et al. [53] applied gamma- tACS while participants completed a phonetic categorization task that also assessed perceptual learning. In five blocks, the participants had to decide whether a syllable represented either /da/ or /ta/. The authors found that gamma-tACS selectively decreased perceptual learning in younger participants. In contrast, participants showed perceptual learning under both control conditions (theta-tACS and no stimulation). In a second study [54], the same authors compared younger to older adults in a similar setting (gamma- vs. theta-tACS). Again, gamma-tACS interfered with younger adults’ performance. On the contrary, older adults’ accuracy increased with gamma-tACS, while with theta-tACS, no effects were observed.

Moliadze and colleagues [52] applied alpha-tACS preceding a phonological decision task. The participants listened to words and decided whether they consisted of two or three syllables. Beta-tACS and sham were used as control conditions. Alpha-tACS significantly increased phonological response speed but also increased error rates compared with sham. There was no difference in either reaction times or error rates between alpha- and beta-tACS.

All included studies applied tACS online and placed electrodes over temporal areas. The stimulation duration varied between 18 and 40 min, with an intensity between 0.8 and 1.7 mA peak-to-peak.

In sum, the results indicated that slow stimulation frequencies in the delta/theta range (3.125–4 Hz) modulated speech perception when directly aligned with speech rhythms. However, for an auditory signal lacking speech rhythm, delta/theta-tACS seemed to benefit the listener’s perception, when it slightly preceded the auditory input. The alignment of the timing of endogenous processes to exogenous auditory events may be associated with oscillatory activity in the delta/theta range that occurred in the auditory cortex [11,87]. This process could possibly be optimized with slow oscillatory tACS. Gamma-tACS selectively disturbed younger adults’ phoneme processing, while it enhanced older adults’ performance. According to the literature, theta and gamma oscillations play an important role in auditory perception [23]. However, there seem to be changes in the importance of gamma-tACS for auditory perception over the lifespan, which warrants further investigation. Alpha-tACS seemed to facilitate phonological response speed but not phonological decision accuracy. Alpha oscillations are related to attention and inhibitory control processes [88,89]. Entraining alpha during phonological processing may interfere with an attention shift on incoming auditory stimuli and thus increase error rates.

### 3.2. Visual Perception

Visual perception is the ability to perceive the environment through the light that enters the eyes. Typical visual perception tasks include the detection or discrimination of visual stimuli, the identification of spatial relationships, and visual rotation tasks.

Brignani et al. [44] stimulated participants at alpha, theta, or gamma frequency during a visual detection and discrimination task. The participants received a visual warning signal after which they had to detect a variously contrasted Gabor patch and discriminate its orientation. Theta- or alpha-tACS decreased performance compared with gamma-tACS and sham. In a study by Gonzalez-Perez et al. [46], participants were stimulated with gamma-tACS before they needed to detect a distractor face or object. They found that gamma-tACS enhanced the detection accuracy for faces and objects compared with sham. Theta-tACS was used as a control condition and had no effects on detection accuracy.

Herring et al. [45] stimulated participants at the individual alpha frequency (IAF) or IAF ± 4 Hz during a visual rotation task. The participants had to indicate the rotation direction of an asterisk in the center of an inward moving high-contrast grating. The authors found that tACS at IAF or IAF ± 4 Hz decreased performance. A different motion detection task was applied in Strüber et al. [47]. They stimulated participants at theta or gamma (in- or anti-phase) during a stroboscopic alternative motion task. The participants had to indicate whether they perceived a motion change in stimuli from vertical to horizontal direction or vice versa. Anti-phase bihemispheric gamma tACS increased the duration of perceived vertical motion. On the contrary, the authors found no effect for in-phase gamma-tACS, in-phase theta-tACS, or anti-phase theta-tACS.

The most common electrode montages for visual perception were mono- or bihemispheric at parieto-occipital or frontal areas. Stimulation duration varied from 5 to 27 min, with an intensity between 1 and 1.5 mA peak-to-peak. Frontal montages more frequently induced phosphenes.

In accordance with previous suggestions of alpha’s role in information gating as well as top-down inhibitory control during visual perception [17,20], alpha-tACS impaired visual perception performance. On the contrary, gamma tACS improved the accuracy of face and object perception. Anti-phase gamma-tACS increased vertical motion perception, while in-phase gamma-tACS had no significant effect. The latter findings reflect the role of gamma oscillations in the bottom-up processing of incoming information. Entrainment to anti-phase gamma-tACS might decelerate the updating process and impair interhemispheric visual motion integration [20]. Theta-tACS was frequently used as a control condition and repeatedly showed no change in visual perception.

### 3.3. Attention

In very general terms, attention is the ability to focus on information. Attention is multifaceted, sometimes being driven by internal goals (i.e., top-down or endogenous attention) and sometimes by external events (i.e., bottom-up or exogenous attention). Typical tasks depend on the type of attention that needs to be evaluated: in tasks of sustained attention, participants need to respond to certain stimuli over a long period of time. In tasks of selective attention, they need to respond to one particular stimulus while ignoring others. In tasks of divided attention, they need to respond to two (or more) stimuli simultaneously. Other attention tasks include the assessment of a shift in attention or the investigation of the attentional blink effect.

Clayton et al. [34] compared alpha-tACS to sham and gamma-tACS during several sustained attention tasks (visual or auditory). Typically, the performance in sustained attention declines with advancing time. Alpha-tACS, however, stabilized performance compared with sham and gamma-tACS. During a very different task, Otsuru et al. [32] also stimulated participants at alpha. They applied a temporal order judgement task which assessed somatosensory processing and sustained attention. In this task, two stimuli were presented bilaterally with varying stimulus onset asynchronies and the participants needed to indicate their temporal order. Alpha-tACS facilitated the ability to discriminate the temporal order of the presented stimuli.

Hopfinger et al. [35] stimulated participants at alpha or gamma during endogenous or exogenous selective spatial attention tasks. Gamma-tACS improved voluntary disengagement from stimuli during the endogenous condition, while alpha-tACS had no reliable effect. Schuhmann et al. [30] also applied spatial attention tasks that tested either the endogenously or the exogenously induced shift in attention, but they stimulated only at alpha frequency. Contrary to Hopfinger et al., they found that alpha-tACS induced an attentional bias towards the left, but only during the endogenous task. Further, Laczó and colleagues [38] applied another spatial attention task. They tested the facilitatory or inhibitory effects of attention on contrast detection thresholds, while stimulating at different gamma frequencies (40, 60, 80 Hz). Only gamma-tACS at 60 Hz improved the participants’ contrast detection performance compared with sham. None of the gamma-tACS frequencies modified the effect of attention on the contrast thresholds.

In a study on selective auditory attention, Deng et al. [31] compared alpha- or theta-tACS with sham. They found disrupted selective attention with alpha-tACS but no effect of theta-tACS. Meier et al. [37] used a multi-site bilateral application of gamma-tACS in another selective attention task (i.e., dichotic listening). While they found no benefit for accuracy, they revealed an advantage for left ear perception with gamma-tACS. Wöstmann et al. [36] also applied a dichotic listening task but stimulated only the left hemisphere with either alpha- or gamma-tACS. Alpha-tACS increased performance for the left-attend condition but decreased performance for the right-attend condition. Gamma-tACS induced the exact opposite.

In two experiments, Yaple and Vakhrushev [33] applied alpha- or beta-tACS (in-phase or anti-phase) during an attentional blink paradigm. They found that anti-phase beta-tACS reduced the attentional blink (i.e., improved detection of the second target).

For attention tasks, electrodes were placed mainly over left, right, or center positions of the parietal cortex. Most studies applied tACS online. Stimulation durations varied from 5 to 55 min, with an intensity between 1 and 2 mA peak-to-peak.

Stimulation with alpha-tACS stabilized or improved sustained attention. On the contrary, alpha-tACS mainly inhibited selective attention performance, while gamma-tACS improved it. During attentional blink tasks, beta-tACS was particularly helpful to reduce the attentional blink. Since alpha plays an important role in the modulation of the focus of attention [89] as well as in visual attention in general [90], these results fit well into the current literature [16,17,20]. Gamma oscillations seem to emerge with the presentation of new external information, while beta seems to appear when information, which is in the current focus of attention, needs to be monitored and when distraction needs to be prohibited [18,20]. Therefore, an interplay between gamma and beta (e.g., via cross-frequency coupling) may be particularly helpful to support attention.

### 3.4. Executive Functions

Executive functions refer to higher level cognitive processes of planning, problem solving, action sequencing, effortful and persistent goal pursuit, inhibition of competing impulses, flexibility in goal selection, and goal conflict resolution. Typical executive function tasks involve judgement tasks, response inhibition tasks (e.g., flanker tasks), or tasks that assess cognitive control.

Kasten et al. [58] stimulated participants at the IAF while participants performed a mental rotation task during which they needed to judge whether two objects were identical or different. They found that the stimulation group performed significantly better (but not faster) in the mental rotation task.

Fusco et al. [60] stimulated during a response inhibition task (i.e., flanker task) where participants had to respond to target letters embedded in a string of distractor letters. Six different blocks of stimulation were applied in randomized order (delta, theta, alpha, beta, gamma, or sham). With theta-tACS, the participants reduced post-error slowing compared with sham—that is, they showed faster responses. In another response inhibition task (i.e., Simon task), van Driel and colleagues [57] stimulated with theta- or alpha-tACS. They found that theta stimulation reduced the relative cost of conflict in low-conflict situations; that is they showed faster responses. Brauer and colleagues [55] stimulated participants at theta during a different response inhibition task (i.e., Go/No-go task). Stimulation had no effect on performance or reaction times.

Wiener and colleagues [59] applied tACS at either alpha or beta frequency during a visual temporal bisection task in which the participants had to categorize the duration of stimuli into “long” or “short”. The authors found faster reaction times for both stimulation frequencies. A second observation was that with beta-tACS, the participants were more likely to classify stimuli as “long” in duration. Reinhart [56] stimulated participants at either in- or antiphase theta frequency during a time estimation task. The participants needed to respond when they had estimated a time-lapse of 1.7 s. With feedback presented at the end of each trial (i.e., “too fast,” “too slow,” or “correct”), they learned to efficiently adapt their response. Right-lateralized in-phase theta-tACS improved response efficiency to the extent that participants no longer needed any feedback. In contrast, right-lateralized anti-phase theta-tACS increased response variability and the participants no longer benefitted from feedback (compared to sham). Repeating the experiment with a left-lateralized in- or antiphase theta-tACS or with gamma-tACS yielded no effects on response efficiency.

For executive functions, all studies applied tACS online. There was more variation in electrode placement than for attention or perception, as electrodes were placed at frontal, fronto-parietal, or parieto-occipital areas. All studies stimulated for 20 min with an intensity between 0.4 and 2 mA peak-to-peak. For fronto-parietal alpha- and beta-tACS, participants reported more visual and somatosensory side effects compared with delta, theta, and gamma.

Theta-tACS as well as alpha-tACS led to increased accuracy and faster responses in a variety of tasks. This is well in line with the suggestion that higher cognitive load is associated with decreased theta and increased alpha and that the level of performance in executive function tasks depends on the interplay between both [15]. Gamma-tACS was typically used as a control condition and did not show any effect on performance. Executive functions rely on top-down control of cognitive processes and gamma has rather been related to bottom-up processing of information [91]. Therefore, the null finding for gamma-tACS seems reasonable. Beta-tACS did not show any consistent effect on performance but it shortened reaction times in one study (and so did alpha-tACS). Beta’s functional role is not clear yet, but it has been proposed to inhibit executive functions [19].

### 3.5. Working Memory

Working memory refers to the capacity to hold information in memory, while performing mental operations on this information. Typical working memory tasks include n-back tasks, digit span (i.e., working memory capacity) tasks, or corsi block span tasks.

In two experiments, Bender et al. [69] applied low (4 Hz) and high theta-tACS (7 Hz) during the completion of a delayed match-to-sample task. During each trial, a different number of colored items (4, 5, or 6) was presented to each hemifield. The participants needed to retain these items and to compare them to a probe item. The authors found an increase in memory capacity with low theta-tACS for those items presented contralateral to the stimulation site.

Polanía et al. [72] stimulated participants at theta frequency (in-phase or anti-phase) while participants performed a letter discrimination task in which they needed to decide whether a test letter matched a previously presented letter or not. The authors found that the in-phase condition led to faster responses compared with both antiphase and sham. The authors applied a control experiment with 35 Hz and confirmed that the results were frequency-specific. In a similar experiment, Alekseichuck and colleagues [68] examined the effect of theta-tACS (in-phase or anti-phase) on a 2-back task. However, in this study, no effect was found for in-phase tACS, while the anti-phase condition led to reduced accuracy and increased response times. Hoy et al. [76] also investigated the effect of tACS on performance in a 2-back task but they applied gamma-tACS. In addition, they applied a 2-back as well as a 3-back task prior to as well as after the stimulation. They found no online effect of tACS but an increase in working memory accuracy for the 3-back condition after the stimulation. For reaction times, neither an online nor an offline effect could be observed.

Violante [73] applied theta-tACS (in-phase or anti-phase) during two different working memory tasks. During a choice reaction task, participants had to indicate the direction of an arrow as accurately and as quickly as possible, while during an n-back task, they had to indicate whether the current digit was the same as one or two times before. The authors found no tACS-related effect on accuracy during the 1- or 2-back task. However, they found shortened reaction times during the choice reaction task for the in-phase condition.

Borghini et al. [78] stimulated participants at alpha frequency during a retro-cued working memory task in younger and older adults. The participants needed to memorize a display of four arrows differing in orientation and color. Following a delay period, one of the four colored arrows reappeared in a random orientation and the participants needed to match it as closely as possible to the original orientation. The authors found that older adults were more likely to respond to targets with alpha-tACS but the effect could not be replicated in a second experiment.

Wolinski et al. [74] used two different electrode montages and applied tACS at different theta frequencies (4 Hz or 7 Hz) during a visual–spatial working memory task. The participants needed to decide whether an array pointed in the same direction as a previous one. For arrays pointing to the left, increased accuracy was found with 4 Hz-tACS, while with 7 Hz-tACS, accuracy decreased. No change in performance was found for arrays pointing to the right. Tseng et al. [77] also applied a visuo-spatial memory task, but they compared gamma-tACS (in-phase or anti-phase) to theta-tACS. While no tACS-related effects were found with in-phase stimulation, gamma-tACS with anti-phase stimulation significantly increased performance in low performers. In another study [71], they applied theta-tACS (in-phase or anti-phase) during a similar visuo-spatial memory task. Comparable to the previous study, only low performers benefitted—in this case, only from in-phase theta-tACS.

Gutteling et al. [75] stimulated either left or right at alpha frequency during a spatial updating paradigm. The participants had to remember a world-fixed target that was briefly flashed prior to a passive displacement of the body and report its remembered location after the motion. The authors found that updating precision was enhanced when a target representation had to be internally remapped to the stimulated hemisphere

Jausovec et al. [70] compared three different electrode montages (left frontal, left parietal, or right parietal) in a between-subject design. They applied tACS at the individual theta frequency followed by three different working memory tasks. During corsi block tapping, the participants needed to repeat the correct order of tapping blocks; during a digit-span task, they needed to recall the correct sequence of numbers, and during a n-back task, they needed to decide whether the current stimulus matched to stimulus one, two, or three times before. For all three tasks, the authors found increased accuracy with theta-tACS, which was more pronounced for the parietal montage.

For working memory, many studies applied tACS online. Comparable to executive functions, electrode montage varied largely and included frontal, fronto-parietal, and parieto-occipital areas. Stimulation duration varied between 6 and 15 min, with an intensity between 1 and 2.25 mA peak-to-peak. Stimulation intensities above 1.5 mA elicited phosphenes when applied close to occipital areas.

Parietal theta-tACS enhanced working memory accuracy, while theta-tACS at frontal areas shortened reaction times. This supports the idea of theta’s involvement in working memory [12] and, more specifically, the hypothesis that theta-band oscillations reflect the organization of sequentially ordered working memory items [92]. Gamma-tACS was particularly beneficial for an increase in working memory accuracy with high cognitive load, in accordance with recent suggestions [93]. Alpha-tACS increased the probability of responding to target stimuli in one study but this effect could not be replicated. Further, alpha-tACS supported the spatial remapping of a target only when it involved the stimulated hemisphere. It has been proposed that alpha-band activity reflects the active inhibition of task-irrelevant information during working memory tasks, which makes alpha-tACS an interesting approach to look into in the future.

### 3.6. Declarative Memory

Declarative memory refers to the ability to retain information about facts or events over a significant period of time as well as to consciously recall the information, usually in response to a specific request to remember. Typical declarative memory tasks are composed of encoding and retrieval (free recall, cued recall, and/or recognition) of words, pictures, or paired stimuli (e.g., face–name, face–occupation).

Nomura et al. [85] stimulated participants at gamma frequency on two consecutive days. On day 1, the stimulation was applied during encoding and on day 2 during recognition (old/new paradigm). Gamma-tACS during recognition improved accuracy but not reaction times compared with sham.

Several studies applied stimulation during paired-associative tasks. De Lara and colleagues [80] applied two cross-frequency stimulations. The experimental setup consisted of four sessions and stimulation was applied on the first and third session during learning of paired-associations. The authors applied three experiments: in the first and second experiment, cross-frequency protocols for tACS where gamma bursts coupled with the peaks or the troughs, or continuously presented during theta wave cycles. They found that short bursts of gamma-tACS coupled with the trough of the theta wave cycle significantly impaired verbal long-term memory performance. Antonenko et al., [79] used theta-tACS during a language-learning paradigm in younger and older adults. During stimulation, the participants were presented with pseudoword-picture pairs in order to learn the association. Later, a recognition task followed (yes/no paradigm). The authors found a significant increase in performance with stimulation in the older group but not in the younger group. Klink et al., [81] stimulated participants at theta frequency during encoding of a face-occupation task. They applied three sessions, in which the participants received transcranial direct current stimulation, tACS, or sham (crossover design). Neither tDCS nor tACS changed memory performance. However, the authors found a significant interaction of age and stimulation method for the retrieval. With increasing age, participants performed significantly worse under sham, but this was not the case for the stimulation groups. Lang et al., [83] also stimulated participants at theta frequency while participants encoded associations. Theta-tACS increased correct recognition performance and reduced forgetting.

Javadi et al., [84] applied gamma-tACS during encoding and retrieval of a list of words in either a congruent or an incongruent condition. In the congruent condition, participants were stimulated at the same frequency during encoding and recognition, while they received different stimulation frequencies in the incongruent condition. Congruent gamma-tACS increased recognition accuracy. Incongruent gamma tACS had no effect on memory performance.

Alekseichuk and colleagues [82] stimulated participants at theta frequency during encoding of faces. They assessed familiarity and recollection as two aspects of long-term memory. The participants had to indicate whether they recognized a face and rate their level of confidence on a six-point Likert scale. Theta-tACS selectively enhanced familiarity. Braun et al. [86] also assessed encoding and recognition of faces (as well as words) but stimulated participants at beta frequency. Beta-tACS had no effect on recognition performance.

As for other complex cognitive functions, electrode montages differed very much and included frontal, temporal, and parietal areas. Most studies stimulated participants during encoding only, while only two studies stimulated participants during encoding and retrieval. Stimulation duration varied between 10 and 20 min, with an intensity between 0.75 and 3 mA peak-to-peak.

In sum, theta-tACS as well as gamma-tACS improved recognition performance for verbal, visual as well as associative stimuli. Beta-tACS had no effect on recognition performance. Theta-tACS interleaved with bursts of gamma-tACS in theta troughs impaired word pair learning. A substantial amount of evidence points towards theta’s role in episodic memory encoding and retrieval [14]. Additionally, gamma and theta oscillations are proposed to interact during encoding and retrieval: in the cortex, gamma oscillations bind perceptual representations, and in the hippocampus, they bind perceptual and contextual information into episodic representations. Theta oscillations temporally order these representations [22]. Beta’s proposed functional role as an inhibitory filter may be important for memory processing; however, future studies need to test this more thoroughly.

### 3.7. Procedural Memory (Motor Learning)

Motor learning refers to processes associated with practice or experience that lead to the acquisition/reacquisition of relatively permanent movement capability. Typical tasks include implicit or explicit motor learning tasks.

Brinkmann et al. [64] used tACS at alpha or beta frequency during a movement selection task. The participants needed to grasp a virtually presented stick either with the left or the right hand. In addition, they needed to indicate verbally which part of the stick (white part, black part) they reached with the thumb as close as possible. The results indicated that the duration from stimulus onset until the verbal response was decreased with alpha-tACS ipsilateral to the requested hand.

Fresnoza and colleagues [63] applied tACS at the IAF or IAF + 2 Hz. Following the application of tACS, younger and older participants completed an implicit motor learning task. The authors found no effect of stimulation on accuracy. However, older adults responded faster with tACS at IAF or IAF + 2 Hz. In younger adults, faster reaction times were found for IAF + 2 Hz only. Giustiniani and colleagues [66] also applied an implicit motor learning task but stimulated participants at gamma frequency. They found that gamma-tACS slowed reaction times during blocks that required retrieval of previously learned sequences. Likewise, Zavecz and colleagues [62] implemented an implicit motor learning task but stimulated participants with theta-tACS. The authors found no tACS-related effects on accuracy or reaction time.

Nowak et al. [65] applied individual beta- or gamma-tACS during a cued reaction time task. The participants were asked to abduct their index finger as soon as possible when a cue appeared. Gamma-tACS improved motor learning performance compared to sham, while beta-tACS had no effects. In Sugata et al. [67], participants also learned a motor sequence after applying three different stimulation frequencies (alpha, beta, or gamma). Gamma-tACS shortened reaction times, while alpha- and beta-tACS had no effects on motor performance.

For motor learning, electrodes were mostly placed at M1 (left). No effect on motor performance was found when the target electrode was placed on the right hemisphere. Both offline and online stimulation were used for procedural memory. Stimulation duration varied between 5 and 30 min, with an intensity between 0.7 and 2 mA peak-to-peak. Adjusting current intensities may be considered when the return electrode is placed over parietal areas to reduce the occurrence of phosphenes.

Performance in the motor domain was mainly modulated by alpha- or beta-tACS. According to the literature, beta seems to be relevant for motor learning as a stronger beta power suppression has been linked to superior learning of a motor sequence [94].

### 3.8. Decision-Making

Decision-making is the cognitive process of choosing between two or more alternatives. Typical decision-making tasks include target–distractor differentiation tasks or preference-based decision-making.

Yaple et al. [61] applied tACS at different frequencies (theta, alpha, beta, or gamma) during two decision tasks with either left or right hemispheric electrode montage. The two tasks comprised an odd/even task (“Is the presented number odd or even”) as well as a high/low task (“Is the presented number higher or lower than five?”). The latter consisted of a gain/loss condition, where participants gained or lost money. The authors found that beta-tACS led to riskier decisions when applied to the left hemisphere. Furthermore, the participants responded faster with left than with right hemispheric montage, regardless of stimulation frequency.

We could only include one study in this domain. The electrodes were placed at frontal areas and the stimulation duration was 40 min with an intensity of 0.5 mA peak-to-peak. Only beta-tACS increased voluntary (risky) decision-making. Frontal beta oscillations have been linked to reward anticipation and unexpected reward feedback [95,96]. In addition, frontal beta activity may indicate communication between cortical and subcortical structures involved in reward processing [96].

### 3.9. Intelligence

Two factors seem to underlie general intelligence: fluid and crystallized intelligence. Fluid intelligence refers to the ability to reason and think flexibly, while crystallized intelligence refers to the accumulation of knowledge, facts, and skills that are acquired throughout the life. Tests of fluid intelligence typically tap reasoning, such as Raven’s Progressive Matrices.

Pahor and Jaušovec [40] applied tACS at IAF + 1 Hz before participants completed Raven’s Progressive Matrices. They found no reliable effect of tACS on performance. Santarnecchi et al. [43] also applied Raven’s Progressive Matrices while stimulating participants at different frequencies (theta, alpha, beta, or gamma). With gamma-tACS, the time that participants needed to solve the most complex matrices decreased significantly. In a follow-up study [42], the same authors applied gamma-tACS while participants again completed Raven’s Progressive Matrices. Theta-tACS and sham were used as control conditions. Comparable to the previous study, gamma-tACS reduced the amount of time that participants needed to solve the most complex matrices, compared with performance under sham and theta-tACS. In addition, they found that low performers benefitted most from gamma-tACS. Neubauer and colleagues [39] also applied Raven’s Progressive Matrices but they stimulated participants with theta-tACS before they solved the task. They found that the participants were able to solve significantly more difficult items with theta-tACS compared with sham, while tACS had no effect on easier items. Grabner et al. [41] examined the influence of alpha- or gamma-tACS during the completion of a verbal intelligence task. The participants were presented with anagrams and asked to rearrange the letters to form meaningful nouns and indicate the correct initial of that noun. Neither gamma-tACS nor alpha-tACS had an effect on performance (compared to sham).

Comparable to other complex cognitive functions, electrode montages were very different and included frontal, parietal, and temporal areas. Stimulation duration varied between 15 and 48 min, with an intensity between 0.75 and 2 mA peak-to-peak.

Online gamma-tACS reduced the amount of time needed to solve the most complex of Raven’s Progressive Matrices, while offline theta-tACS increased accuracy on these complex items. In addition, initially low performers benefitted most from stimulation. Neither gamma- nor theta-tACS had an effect on solving easier items. Interestingly, online gamma-tACS had no effect on a verbal intelligence task, which examined crystalline intelligence. Thus, gamma-tACS may be particularly beneficial for fluid intelligence tasks. In general, fluid intelligence has been associated with theta-gamma coupling [97]; thus, combining the beneficial effects of both frequencies by applying cross-frequency modulation may optimize tACS effects on fluid intelligence.

## 4. Discussion

This review summarized how tACS at varying frequencies modulated cognitive performance. Figure 4 gives an overview of the frequencies that were used during stimulation. It appeared that alpha, gamma, and theta have been used very frequently, while beta and delta were less commonly applied.

The cognitive domains described in this review differ greatly in complexity. Auditory and visual perception, for example, are basic cognitive functions that often serve as a prerequisite for other, more complex functions. For these basic cognitive functions, this review revealed that online stimulation (see Figure 1E) with entrainment (see Figure 1D) of slow oscillations modulated the perception of speech rhythm [48,49,50,51], while entrainment of gamma oscillations exerted an age-dependent effect by improving phonological discrimination only in elderly participants [53,54]. On the contrary, alpha-tACS impaired visual and phonological discrimination performance [44,45]. Some frequencies led to no performance changes with online stimulation (i.e., theta- and beta-tACS) [52,54]. Online stimulation aiming to influence phase coherence (see Figure 1C; here: anti-phase, bi-hemispheric gamma-tACS) impaired visual motion change detection. On the contrary, in-phase gamma- or theta-tACS had no reliable effect [47]. Offline stimulation (Figure 1E) with gamma-tACS improved visual discrimination performance, probably reflecting a successful and longer lasting amplitude increase in underlying gamma oscillations (see Figure 1A) [46].

In the real world, slow oscillatory tACS may be a useful tool to support processes of brain plasticity after the implantation of a cochlear device [98]. The benefit of such a device crucially depends on the brain’s ability to learn to classify neural activity evoked by the cochlear implant. Here, slow oscillatory tACS may be particularly helpful (see also [99] for the clinical potential of electrical stimulation).

Attention, executive functions, and working memory are essential cognitive functions required for higher-order processes. The studies on attention described in this review used visual or auditory stimuli and applied online stimulation, mostly with the goal of entrainment (Figure 1D). Performance was successfully improved with alpha-tACS, which modulated sustained visual attention [32,34] as well as the detection of auditory left-attend stimuli [36]. In addition, gamma-tACS improved the detection of right-attend stimuli [36] as well as endogenous selective visual attention [30,35,38]. In contrast, alpha-tACS decreased selective visual [30,35,38] and auditory attention [31,37]. Theta- and gamma-tACS showed no effect on selective auditory attention [31,37]. Only one study aimed for phase coherence effects: anti-phase beta-tACS reduced the attentional blink [33].

For executive functions, most studies applied online stimulation with the goal of entrainment. Faster responseswere induced by theta-tACS during response inhibition [55,57,60] and by alpha- or beta-tACS during time estimation [59]. Additionally, individual alpha-peak-tACS increased mental rotation performance [58]. Delta-, alpha-, beta-, and gamma-tACS exerted no effect on response inhibition [55,57,60] and gamma-tACS showed no effect on time estimation [56]. One study using online stimulation to affect phase coherence showed that right-lateralized in-phase theta-tACS increased time estimation accuracy, while anti-phase theta-tACS had no reliable effect [56].

For working memory, only one study used offline stimulation: gamma-tACS improved n-back task performance [76]. For online studies aiming for entrainment, tACS at the individual theta-peak improved n-back performance [70] and low-theta-tACS improved visual spatial working memory [74]. In addition, alpha-tACS improved spatial updating for the stimulated hemisphere [75] and low-theta-tACS during a delayed-match-to-sample task improved accuracy for stimuli processed with the contralateral hemisphere [69]. On the contrary, gamma-tACS had no effect on letter discrimination [72] or n-back task performance [76]. Online studies with the goal of affecting phase coherence showed that low performers benefited from anti-phase gamma- or in-phase theta-tACS [71,77]. Likewise, in-phase theta-tACS quickened response times on letter discrimination and choice reaction tasks [72,73]. In contrast, anti-phase theta-tACS impaired n-back accuracy [68].

Declarative and procedural memory are complex cognitive functions that rely on basic cognitive functions (e.g., perception, attention). Studies that applied online stimulation during encoding found that theta-tACS increased familiarity-based recognition performance in young [82,83] and older adults [79], while it improved cued recall performance only in the oldest participants [81]. Applying gamma-tACS online during encoding and retrieval improved memory performance [84,85], but beta-tACS led to no such effect [86]. One study investigated the effect of cross-frequency coupling (Figure 1B): When gamma-tACS was interleaved with theta-tACS troughs during memory encoding, retrieval accuracy was impaired, while gamma-tACS bursts during theta-tACS peaks induced no reliable effect [80].

All studies investigating tACS effects on motor learning applied stimulation online. Individual alpha-peak (IAF) tACS improved older adults’ implicit motor learning performance and alpha-tACS at IAF + 2 Hz improved performance in younger and older adults. On the contrary, online gamma-tACS slowed reaction times, while online theta-tACS had no effect on implicit motor learning [62,63,66]. Gamma-tACS increased motor sequence learning, while alpha- and beta-tACS had no effect [65,67]. Alpha-tACS increased response time on movement selection, while beta-tACS had no effect [64].

Decision-making and fluid intelligence are also highly complex cognitive functions. For decision-making, all studies applied stimulation online and aimed to entrain specific frequencies: beta-tACS induced riskier decisions, while left-lateralized theta-, alpha-, and gamma-tACS only led to faster responses [61]. For fluid intelligence, online (but not offline) gamma-tACS reduced the time needed to solve difficult fluid intelligence problems. On the contrary, offline (but not online) theta-tACS increased the likelihood of solving difficult fluid intelligence problems [39,42,43]. For crystalline intelligence, neither alpha- nor gamma-tACS showed a reliable effect when applied online [41].

In sum, we found that theta-tACS was beneficial for several cognitive functions, including working memory, executive functions, and declarative memory. Gamma-tACS enhanced both auditory and visual perception but did not change performance in tasks of executive functions. For attention, the results were less consistent but indicated that with alpha- (sustained attention) or gamma-tACS (selective attention), performance was improved. A comparison of online vs. offline effects of tACS was only possible for some of the included cognitive domains. Gamma-tACS was more effective for improving visual discrimination performance [46] and n-back task performance [76] when applied offline. For fluid intelligence, online gamma-tACS improved performance particularly, while theta-tACS was only effective when applied offline [39,42,43]. In general, more studies directly comparing different frequencies are needed. However, this may be difficult to implement because the duration of the study would increase, which could lead to fatigue in the participants. Due to the heterogeneity in the tACS protocols of this review, it is difficult to recommend any particular stimulation parameters. Instead, we would like to encourage researchers to follow the practical guidelines proposed by Antal and Hermann when designing their studies [2]. One thing that we would like to note is that the occurrence of phosphenes can be reduced by applying an individual current intensity threshold [7].

Although the results of our review are promising, we believe that some considerations need to be taken into account before a transfer of tACS from the laboratory to the real-world would be possible. Real-world situations are typically dynamic, complex, and difficult to control. In contrast, experiments in the laboratory are usually well-controlled and simplified, with artificial stimuli. Due to the complexity and dynamics of real-world situations, a breakdown of the individual contribution of each cognitive domain to performance in a cognitive task seems difficult (e.g., for successful memory formation, attention and executive functions do play a role). In addition, real-world situations may be noisier, with many distractors, so focusing on a particular stimulus may be more challenging than in the laboratory. At an intermediate point on the spectrum ranging from approaches with maximal control to those with maximal behavioral relevance, real-world-like laboratory experiments test cognitive functions in settings resembling everyday situations. Specifically, they constrain the range of potential processes present during the experiment by also using process-specific tasks adapted so that the noise characterizing real-world situations is still present, albeit in a more controlled fashion. This intermediate approach allows the study of cognitive functions by scaling up from the simplest to more complex settings that capture some aspects of the complexity of the real-world (e.g., real-world objects instead of colored squares) while still remaining in the realm of well-controlled stimuli and well-understood tasks.

Several attempts have been made for translating laboratory experiments to the real-world. For episodic memory, the What-Where-When task [100] examines spatial memory as well as object memory, by requiring participants to hide 16 objects at different locations and, after a delay, to retrieve were the objects were hidden. Performance in this task correlated well with laboratory episodic memory abilities but provided more ecological validity. In such an experiment, tACS could be applied at retrieval rather than encoding because, during encoding, the participant will move through a location (e.g., an office) to hide the different objects. This would allow us to test whether tACS would be able to improve retrieval.

Attention and working memory assessments can be translated into settings that are more realistic too. One study investigated whether working memory-driven attentional capture could also be found in a real-world visual search with stimuli taken from the IKEA^®^ catalogue [101]. Another line of research has addressed the question of how attention operates in realistic viewing conditions. One study used a very large (>1000) set of object and action categories to investigate how attention modulates the representational semantic space. The stimuli were presented in a movie clip while participants were looking for either people or vehicles [102]. Others have found that, with real-world objects, participants do not show a working memory capacity limit, in contrast to the fixed capacity that is observed with simple stimuli [103]. Still others found that, when participants briefly looked at a group of faces or a crowd of people, they quickly extracted facial expressions, gender, as well as the mean head rotation of the group, e.g., [104].

For other cognitive domains (e.g., visual or auditory perception), a transfer to real-world applications might be more challenging. For perception, there are several differences between laboratory and real-life applications that may cause behavioral differences that are rather due to the experimental approach than to behavior. First, participants are typically immobile in the lab, since head movements are prevented with a forehead and chin rest. Second, aside from the respective visual or auditory stimuli, no other sensory input is given that would normally be integrated with the respective modality. Third, no interaction with the environment is possible in the laboratory. Virtual reality may be a step towards a transfer from the laboratory to the real-world.

In any case, besides the behavioral assessments of tACS effects outside the lab, it may be particularly interesting to simultaneously apply electrophysiological recordings via EEG. During recent years, mobile EEG equipment has been developed that allows recordings of human brain activity outside the lab using small, wireless hardware and electrode arrays designed to minimize interference and the discomfort of the equipment for participants [105,106,107]. These systems have been used outside the laboratory to study human brain oscillations during different cognitive tasks, such as the formation of episodic memories [108], sensory motor activity during walking [109], or spatial navigation [110], and they may be well suited to assessing the effects of tACS on brain oscillations in naturalistic environments. This will be a crucial step for translating brain stimulation into real-world settings.

While tACS has been repeatedly shown to modulate human brain oscillations in the lab, there is evidence that its effects may strongly depend on brain states [111,112] and external settings [113]. It is thus crucial to validate that tACS effects can be reliably induced in less controlled environments outside the laboratory and to find out which environmental conditions may determine stimulation success. While the assessment of online effects of tACS on EEG oscillations remains a major challenge due to a massive stimulation artefact introduced to electrophysiological recordings—see [114] for a review—the study of stimulation aftereffects in the EEG still provides the most direct evidence for tACS efficacy in naturalistic, or real-life, experiments. In addition, EEG-guided tACS parameter selection may be useful to increase the effects of stimulation. It has been shown that the relation between tACS frequency and the dominant frequency in the brain can influence stimulation effects, along with anatomical differences, causing variations in the electric fields [6,115]. Individualizing stimulation frequencies and montages based on EEG recordings prior to stimulation may thus help to increase the reliability of tACS. For the latter, electric field simulations with advanced electrode optimization algorithms are increasingly available [116,117,118,119].

We would like to note that more studies should examine the long-term effects of tACS on cognitive performance. Changes in oscillatory activity beyond the time of stimulation have been reported very frequently (most notably for alpha-tACS) but long-term effects on cognitive performance still need to be determined. We believe that these may help to design and conduct effective real-world paradigms. Besides controlling stimulation parameters, the influence of the menstrual cycle or hormones in general may be considered when planning tACS studies as these variables will likely influence the results.

Our review might be limited by the predefined minimum sample size of studies to be included. A sample size of *n* = 20 (or *n* = 15 in the case of a within-subject design) might not achieve enough power to detect a true effect. However, the required sample size in a study depends on the reliability of the dependent measure, the design, and the statistical analysis plan, which is why the decision on a minimum sample size for an inclusion in a systematic review is challenging. Although some of the included studies reported a priori sample size calculations adjusted to their particular statistical analysis plan, we would like to encourage future studies to do so more regularly. In addition, we would like to encourage researchers to follow the recommendations of Button et al. [120] to improve reproducibility in neuroscience.

This systematic review summarized laboratory studies that applied tACS at varying frequencies to modulate cognitive performance. The successful transfer of tACS from the laboratory to the real-world first requires innovative approaches to validate tACS effects in less controlled environments. If successful, tACS may become a powerful tool to modulate everyday cognitive performance.

## Figures and Tables

**Figure 1 brainsci-10-00932-f001:**
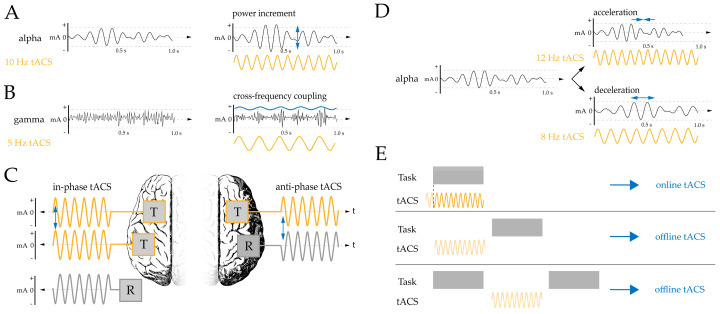
Overview of different ways to modulate brain oscillations (**A**–**D**). (**E**) Online tACS: applied during a cognitive task; offline tACS: applied either prior to a cognitive task or between tasks. Note: T, target electrode; R, return electrode; mA, milliamps; t, time.

**Figure 2 brainsci-10-00932-f002:**
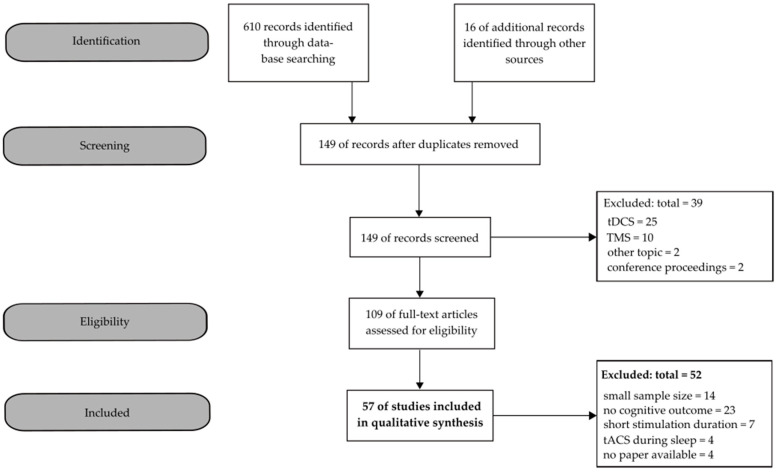
Flow chart of the identification and inclusion of studies in the current systematic review.

**Figure 3 brainsci-10-00932-f003:**
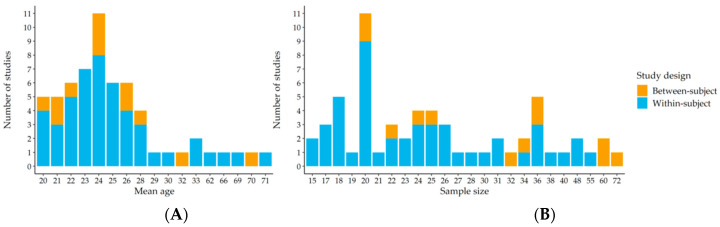
Histograms of the mean age (**A**) as well as the mean sample sizes (**B**) of all included studies.

**Figure 4 brainsci-10-00932-f004:**
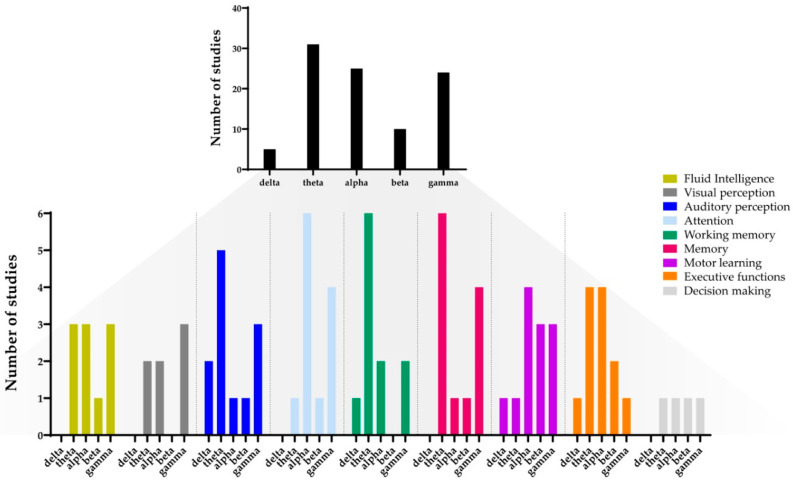
Number of times tACS with a certain frequency was applied in all included studies (**top**) or in each cognitive domain (**bottom**).

**Table 1 brainsci-10-00932-t001:** Overview of study designs, tasks, stimulation parameters, electrode positions, as well as risk of bias for all included studies.

Authors	Participants and Design	Task and Dependent Variables	Electrode Montage and Device	Stimulation Parameters	Outcome/Risk Bias
**Attention**					
**Alpha**					
[30]	*N* = 36 (mean age: 21.56 years, range 18–29 years) Crossover within-subject design (tACS vs. sham)	Endogenous and exogenous spatial attention task *Reaction time bias score*	Left parietal cortex (P3) + centered above target electrode *Device*: high-definition tES (NeuroConn, Ilmenau, Germany)	*Frequency*: 10 Hz/sham *Intensity*: 1 mA peak-to-peak *Duration*: 35–40 min *Electrode size*: target electrode (round-shaped, diameter: 2.1 cm), return electrode (round-shaped, outer diameter: 11 cm, inner diameter: 9 cm) *Timing*: online	↗ (for endogenous task only) Low risk
[31]	**Exp. 1** *N* = 20 (mean age: 26.6 ± 4.1 years) **Exp. 2** *N* = 18 (mean age: 22.11 ± 2.4 years) **All Exp.** Crossover within-subject design (tACS vs. sham)	Selective auditory attention task *Number of /ga/-syllables in spatial (left or right direction) or non-spatial trials (female/male speakers)*	Parietal cortex (P2) + (CP2/P4/Pz/PO4) Soterix M × N 9 high definition-tES Stimulator (Soterix Medical, New York, NY)	*Frequency*: **Exp. 1** 10 Hz/sham; **Exp. 2** 6 Hz/sham *Intensity*: 1.5 mA peak-to-peak *Duration*: 20 min incl. 30 s ramp-in/-out *Electrode Size*: not provided *Timing*: online	↘ (with alpha-tACS) − (with theta-tACS) Low risk
[32]	**Exp. 1** *N* = 24 (mean age: 21) **Exp. 2** *N* = 12 (mean age: 21.2 ± 0.2 years) **All Exp.** Crossover within-subject design (tACS vs. sham)	Temporal order judgement task *Forced-choice verbal report of stimuli order*	**Exp. 1** Left somatosensory cortex (C3) or left posterior parietal cortex (P3) + ipsilateral lower cheek **Exp. 2** Right posterior parietal cortex (P4) + ipsilateral lower cheek *Device*: Eldith (NeuroConn GmbH, Ilmenau, Germany)	*Frequency*: 10 Hz/sham *Intensity*: 1 mA peak-to-peak *Duration*: 270 s *Electrode size*: both 5 × 5 cm *Timing*: online	↗ Low risk
**Alpha and Beta**					
[33]	**Exp. 1** *N* = 18 (mean age: 20.66 ± 2.53 years) Crossover within-subject design (10 Hz versus 20 Hz versus sham) **Exp. 2** *N* = 15 (mean age: 20.26 ± 2.35 years) Crossover within-subject design (sham vs. 10 Hz in-phase vs. 10 Hz anti-phase vs. 20 Hz in-phase vs. 20 Hz anti-phase)	Attentional blink task *Detection of second target stimuli*	**Exp. 1** Right posterior parietal cortex (P4) + right deltoid **Exp. 2** Right posterior parietal cortex (P4) and left frontal cortex (F3) + right deltoid *Device*: **Exp. 1** BrainStim (EMS Medical, Bologna, Italy); **Exp. 2** StarStim 8 (Neuroelectrics, Boston, Massachusetts)	*Frequency*: **Exp. 1** 10 Hz/20 Hz/sham; **Exp. 2** 10 Hz in-phase/10 Hz anti-phase/20 Hz in-phase/20 Hz anti-phase/sham *Intensity*: 0.35 mA *Duration*: **Exp. 1** 35 min; **Exp. 2** 50 min *Electrode size*: **Exp. 1** both 7 × 5 cm; **Exp. 2** both round-shaped electrodes (25 cm^2^) *Timing*: online	↗ (with anti-phase beta-tACS) − (with alpha-tACS or in-phase beta-tACS) Low risk
**Alpha and Gamma**					
[34]	**Exp. 1** *N* = 48 (mean age: 22.7 ± 2.9 years) **Exp. 2** *N* = 37 (mean age: 22.7 ± 3.5 years) **Exp. 3** *N* = 41 (mean age: 23.2 ± 2.7 years) **Exp. 4** *N* = 43 (mean age: 23.1 ± 3.1 years) Crossover within-subject design (tACS vs. sham vs. control (50 Hz) tACS)	Visual or auditory sustained attention tasks *Correct responses, reaction times*	Occipital cortex (Oz) + vertex (Cz) *Device*: Starstim (Neuroelectrics, Barcelona)	*Frequency*: 10 Hz/50 Hz/sham *Intensity*: 2 mA peak-to-peak *Duration*: 11 min *Electrode size*: both 25 cm^2^ (round-shaped) *Timing*: online	↗ (with alpha-tACS) − (with gamma-tACS) Low risk
[35]	*N* = 23 (age range: 18–27 years) Crossover within-subject design (alpha tACS vs. gamma tACS vs. sham)	Selective spatial attention task (endogenous or exogenous) *Reaction times*	Inferior parietal lobe (P6) + vertex (Cz) *Device*: NeuroConn DC Stimulator Plus (NeuroCare, Ilmenau, Germany)	*Frequency*: 10 Hz/40 Hz/sham *Intensity*: 2 mA peak-to-peak *Duration*: Endogenous blocks: ~33 min Exogenous blocks: ~21 min *Electrode size*: target electrode 5 × 5 cm, return electrode 5 × 7 cm *Timing*: online	↗ (with alpha-tACS) − (with gamma-tACS) Low risk
[36]	*N* = 20 (age range: 19–31 years) Crossover within-subject design (tACS vs. control vs. sham)	Dichotic Listening task *Correct responses*	Fronto-central and temporo-parietal cortex (FC5) + TP7 *Device*: NeuroConn DC Stimulator Plus (NeuroCare, Ilmenau, Germany)	*Frequency*: 10 Hz/47.1 Hz/sham *Intensity*: 1 mA peak-to-peak *Duration*: 25 min *Electrode Size*: both 7 cm^2^ (round-shaped, 3 cm in diameter) *Timing*: online	alpha-tACS: ↗ (for left-attend condition) ↘ (for right-attend condition) gamma-tACS: ↘ (for left-attend condition) ↗ (for right-attend condition) Low risk
[37]	*N* = 26 (mean age: 28.5 ± 7.9 years) Crossover within-subject design (tACS vs. sham)	Dichotic Listening task *Correct responses*	Multifocal approach: (FT9 + FT10/F3 + F4/CP5 + CP6/P5 + P6) *Device*: NeuroConn DC Stimulator Plus (NeuroCare, Ilmenau, Germany)	*Frequency*: 40 Hz/sham *Intensity*: 1 mA peak-to-peak *Duration*: 20 min *Electrode Size*: all electrodes round-shaped, 12 mm in diameter *Timing*: online	↗ (for left ear-perception only) Low risk
**Gamma**					
[38]	*N* = 20 (mean age: 25.8 ± 6.2) Crossover within-subject design (40 Hz vs. 60 Hz vs. 80 Hz vs. sham)	Visual spatial attention task *Correct responses*	Occipital area (Oz) + vertex (Cz) *Device*: DC-Stimulator Plus (NeuroConn, Ilmenau, Germany)	*Frequency*: 40 Hz/60 Hz/80 Hz/sham *Intensity*: 1.5 mA peak-to-peak *Duration*: **Exp. 1**: 45 ± 10 min, **Exp. 2**: 15 ± 5 min *Electrode size*: target electrode 4 × 4 cm, return electrode 7 × 4 cm *Timing*: online	↗ (with 60 Hz gamma-tACS) − (with other tACS) Low risk
**Intelligence**					
**Theta**					
[39]	*N* = 20 (mean age: 24.9 ± 3.3 years) Crossover within-subject design (tACS vs. sham)	Raven’s progressive matrices *Correct responses*	left temporal cortex (T7) + C3 *Device*: DC-Stimulator Plus (NeuroConn, Ilmenau, Germany)	*Frequency*: 5 Hz/sham *Intensity*: 1.5 mA peak-to-peak *Duration*: 15 min *Electrode size*: both 5 × 7 cm *Timing*: offline	↗ (for difficult items only) Low risk
**Alpha**					
[40]	*N* = 20 (mean age: 20.18 ± 0.40) Crossover within-subject design (tACS vs. sham)	Raven’s progressive matrices *Correct responses*	Frontal cortex (F3) + F4 *Device*: NeuroConn DC Stimulator Plus (NeuroCare, Ilmenau, Germany)	*Frequency*: 1 Hz above IAF/sham *Intensity*: Modus = 1.75 mA peak-to-peak (Range = 1 mA to 2 mA) *Duration*: 15 min *Electrode Size*: both 7 × 5 cm *Timing*: offline	− Low risk
**Alpha and Gamma**					
[41]	*N* = 22 (mean age: 23.00 ± 2.85 years) Crossover within-subject design (alpha tACS vs. gamma tACS vs. sham)	Verbal intelligence task *Correct responses*	Frontal cortex (F3) + F4 *Device*: Eldith (NeuroConn GmbH, Ilmenau, Germany)	*Frequency*: 10 Hz/40 Hz/sham *Intensity*: 1 mA peak-to-peak *Duration*: 30 min incl. 100 cycles ramp-in/-out *Electrode size*: both 5 × 7 cm *Timing*: online	− Low risk
**Gamma and Theta**					
[42]	**Exp. 1** *N* = 24 (mean age: 23.8 ± 3.14 years) **Exp. 2** *N* = 34 (mean age: 24.3 ± 2.76 years) Crossover within-subject design (tACS vs. sham vs. control)	Modified version of Raven’s progressive matrices *Correct responses, response times*	Frontal cortex (left MFG) + vertex (Cz) *Device*: Eldith DC-Stimulator, NeuroConn, Germany	*Frequency*: **Exp. 1** 40 Hz/ 5 Hz/sham; **Exp. 2** 40 Hz/sham *Intensity*: 0.75 mA peak-to-peak *Duration*: ~30 min *Electrode Size*: both 5 × 5 cm *Timing*: online	↗ (with gamma-tACS, benefit for low performer only) − (with other tACS) Low risk
**Multiple Frequencies**					
[43]	*N* = 20 (mean age: 20.2 ± 12.3 years) Crossover within-subject design (tACS vs. sham vs. control)	Raven’s progressive matrices *Correct responses, response times*	Frontal cortex (left MFG) + vertex (Cz) *Device*: Eldith DC-Stimulator, NeuroConn, Germany	*Frequency*: 5 Hz/10 Hz/20 Hz/40 Hz/sham *Intensity*: 0.75 mA peak-to-peak *Duration*: through task (~48 min), per condition *Electrode Size*: both 5 × 7 cm *Timing*: online	↗ (with gamma-tACS) − (with other tACS) Low risk
**Visual Perception**					
**Alpha**					
[44]	*N* = 96, divided into 4 groups with each *N* = 24 6 Hz (mean age: 21 ± 2.4 years) 10 Hz (mean age: 21 ± 3.2 years) 25 Hz (mean age: 22 ± 2.1 years) sham (mean age: 22 ± 2.3 years) Between-subject design	Visual detection and discrimination task *Correct responses*	Parieto-occipital cortex (PO7/PO8) + vertex (Cz) *Device*: NeuroConn DC Stimulator (NeuroCare, Ilmenau, Germany)	*Frequency*: 6 Hz/10 Hz/25 Hz/sham *Intensity*: 1 mA peak-to-peak *Duration*: 5 min *Electrode size*: target electrode 16 cm^2^, return electrode 35 cm^2^ *Timing*: online	↘ (with theta- or alpha-tACS) Low risk
[45]	*N* = 15 (mean age: 24.3 ± 0.7 years) Crossover within-subject design	Visual rotation detection task *Correct responses*	Visual cortex stimulation montage (Oz) + Cz Retinal stimulation montage (Fpz) + Cz *Device*: NeuroConn DC Stimulator Plus (NeuroCare, Ilmenau, Germany)	*Frequency*: IAF/IAF+4 Hz/IAF–4 Hz *Intensity*: Oz–Cz: 0.96 ± 0.3; Fpz–Cz: 2.31 ± 1.1 mA peak-to-peak *Duration*: 27 min (~5.4 s per frequency-montage combination) *Electrode size*: both 5 × 5 cm *Timing*: online	↘ Low risk
**Gamma**					
[46]	*N* = 60 (mean age: 32.22 ± 7.96 years) 40 Hz (*N* = 12) 40 Hz (*N* = 16) 5 Hz (*N* = 16) sham (*N* = 16) Between-subject design	Face perception task, object perception task, *Correct responses, reaction times*	Occipital cortex (PO8) + frontal cortex (FP1) *Device*: Neuroelectrics, Barcelona, Spain	*Frequency*: 40 Hz/5 Hz/sham *Intensity*: 1.5 mA peak-to-peak *Duration*: 20 min *Electrode size*: both 25 cm^2^ *Timing*: offline	↗ (with gamma-tACS) − (with theta-tACS) Low risk
[47]	*N* = 45 (mean age: 24.9 ± 4.1) **Exp. 1** *N* = 17 **Exp. 2** *N* = 13 **Exp. 3** *N* = 15 **All Exp.** Crossover within-subject design	Stroboscopic alternative motion task *Motion dominance index (% duration of perceived motion direction)*	**Exp. 1** and **Exp. 2**: parieto-occipital cortex (P7-PO7) + (P8-PO8) **Exp. 3**: Central (C3/C4) + occipital areas (O1/O2) *Device*: NeuroConn, Ilmenau, Germany	*Frequency*: **Exp. 1** anti-phase 40 Hz tACS/sham; **Exp. 2** antiphase 6 Hz/sham; **Exp. 3** in-phase 6 Hz/40 Hz/sham *Intensity*: 0.5–1.5 mA peak-to-peak *Duration*: 15 min *Electrode size*: **Exp. 1** and **Exp. 2**: both 5 × 7 cm; **Exp. 3**: both 3.9 × 3.9 cm *Timing*: online	↘ (with anti-phase gamma-tACS) − (with other tACS) Low risk
**Auditory Perception**					
**Delta/Theta**					
[48]	*N* = 17 (mean age: 33 ± 8 years) Crossover within-subject design	Word detection task *Correct responses*	Left temporal cortex (T7) + C3 *Device*: DC-Stimulator MR (NeuroConn, Ilmenau, Germany)	*Frequency*: 3.125 Hz/sham *Intensity*: 1.7 mA peak-to-peak *Duration*: ~30 min *Electrode size*: target electrode 3 × 3 cm, return electrode 5 × 7 cm *Timing*: online	− High risk (small sample sizes)
[49]	**Exp. 1** *N* = 27 (mean age: 31 ± 7 years) **Exp. 2** *N* = 19 (mean age: 21 ± 2 years) Crossover within-subject design	Speech perception task *Correct responses*	**Exp. 1** *Unilateral montage*: left temporal cortex (T7) + C3 **Exp. 2** *Bilateral montage*: temporal cortex (T7/T8) + 2 large ring electrodes (around target electrode) *Device*: NeuroConn DC Stimulator MR (NeuroCare, Ilmenau, Germany)	*Frequency*: 3.125 Hz/sham *Intensity*: **Exp. 1** 1.2 mA peak-to-peak; **Exp. 2** 1.7 mA peak-to-peak *Duration*: ~30 min *Electrode Size*: both 5 × 7 cm *Timing*: online	Bilateral stimulation: ↗ (when stimulation and speech were aligned) ↘ (when stimulation and speech were not aligned) − (unilateral stimulation) Low risk for **Exp. 1** High risk for **Exp. 2** (small sample size)
**Theta**					
[50]	*N* = 20 (age range: 18–38 years) Crossover within-subject design	Stream perception task *Correct responses*	Temporal cortex (T7/T8) + L/R side of midline (Cz) *Device*: NeuroConn DC Stimulator (NeuroCare, Ilmenau, Germany)	*Frequency*: 4 Hz/sham *Intensity*: 0.8 ± 0.1 mA peak-to-peak *Duration*: 40 min *Electrode Size*: both target electrodes 5 × 5, both return electrodes 5 × 7 cm *Timing*: online	↗ (when streams were synchronous with entrained oscillatory activity) Low risk
[51]	*N* = 22 (age range: 18–28 years) Crossover within-subject design	Speech perception task *Correct responses*	Temporal cortex (T7/T8) + Cz *Device*: DC-Stimulator MR (NeuroConn, Ilmenau, Germany)	*Frequency*: 4 Hz/sham *Intensity*: **Exp. 1**: 0.9 ± 0.1 mA peak-to-peak, **Exp. 2**: 1 ± 0.1 mA peak-to-peak *Duration*: 36 min *Electrode size*: both target electrodes 5 × 5 cm, return electrode 10 × 7 cm *Timing*: online	↗ Low risk
**Alpha and Beta**					
[52]	*N* = 26 (mean age: 21.4 ± 4.7 years) Crossover within-subject design	Phonological decision task *Correct responses, reaction times*	Fronto-temporal cortex (Crossing point of T3-Fz × F7-Cz) + (T4-Fz × F8-Cz) *Device*: NeuroConn DC Stimulator (NeuroCare, Ilmenau, Germany)	*Frequency*: 10 Hz/16.18 Hz/sham *Intensity*: 1 mA peak-to-peak *Duration*: 20 min *Electrode Size*: both 9 cm^2^ (round-shaped) *Timing*: offline	With alpha-tACS: ↗ (reaction time) ↘ (error rates) − (with other tACS) Low risk
**Gamma and Theta**					
[53]	**Exp. 1** *N* = 21 (mean age: 24.3 ± 2.0 years) **Exp. 2** (control) *N* = 17 (mean age: 27.5 ± 3.3 years) **All Exp.** Crossover within-subject design	Phonetic categorization task *Correct responses, perceptual learning*	**Exp. 1** only temporal areas (T7) + T8 *Device*: NeuroConn DC Stimulator (NeuroCare, Ilmenau, Germany)	*Frequency*: **Exp. 1** 40 Hz/6 Hz; **Exp. 2** No stimulation *Intensity*: 1 mA peak-to-peak *Duration*: 18 min in total, in 6 min blocks *Electrode Size*: both 5 × 7 cm *Timing*: online	↗ (with gamma-tACS in older adults) ↘ (with gamma-tACS in younger adults) − (with theta-tACS) Low risk
[54]	*N* = 25 (mean age: 24.1 ± 2.24 years) *N* = 20 (mean age: 69.8 ± 4.4 years) Between-subject/Crossover design	Phonetic categorization task *Correct responses, perceptual learning*	Temporal cortex (T7) + T8 *Device*: NeuroConn DC Stimulator (NeuroCare, Ilmenau, Germany)	*Frequency*: 40 Hz/6 Hz *Intensity*: 1 mA peak-to-peak *Duration*: 10 min *Electrode Size*: both 5 × 7 cm *Timing*: online	↘ (with gamma-tACS) ↗ (with theta-tACS) Low risk
**Executive functions**					
**Theta**					
[55]	*N* = 23 (mean age: 22.91 years, range 18–30) Crossover within-subject design	Go/NoGo task *Correct responses, reaction times*	Fronto-temporal area (T4-Fz × F8-Cz) + supraorbital *Device*: NeuroConn DC Stimulator (NeuroCare, Ilmenau, Germany)	*Frequency*: 6 Hz/sham *Intensity*: 1 mA peak-to-peak *Duration*: 20 min *Electrode Size*: both 5 × 5 cm *Timing*: online/offline	− Low risk
[56]	*N* = 90 (*N* = 30 for each experiment) **Exp. 1** mean age: 26 years **Exp. 2** mean age: 27 years **Exp. 3** mean age: 26 years Crossover within-subject design	Time estimation task *Correct responses*	**Exp. 1/3** medial frontal and lateral prefrontal cortex (right: MFC/IPFC) + 4 surrounding return electrodes **Exp. 2** similar to **Exp. 1**, but left hemisphere *Device*: M x N 9 high-definition Stimulator (Model 9002A, Soterix Medical, New York, NY)	*Frequency*: 6 Hz in-phase/anti-phase/35 Hz/sham *Intensity*: 1 mA peak-to-peak *Duration*: **Exp. 1/2**: 20 min; **Exp. 3**: 20 min *Electrode size*: high definition-tACS *Timing*: online	↗ (tACS on right hemisphere) − (tACS on left hemisphere or gamma-tACS) Low risk
**Theta and Alpha**					
[57]	*N* = 31 (age range: 19-31 years) Crossover within-subject design	Simon task *Correct responses, response times*	left frontal cortex (between FCz and Cz) + both cheeks *Device*: DC-Stimulator MR (NeuroConn, Ilmenau, Germany)	*Frequency*: ITF, IAF *Intensity*: 2 mA peak-to-peak *Duration*: ~20 min *Electrode size*: target electrode 9 cm^2^, both return electrodes 35 cm^2^ *Timing*: online	↗ (with theta-tACS) − (with alpha-tACS) Low risk
**Alpha**					
[58]	*N* = 20 (mean age: 26 ± 3 years) *tACS* (*N* = 10) *sham* (*N* = 10) Between-subject design	Mental rotation task *Correct responses, reaction times*	Occipital cortex (Oz) + vertex (Cz) *Device*: NeuroConn (NeuroCare, Ilmenau, Germany)	*Frequency*: 10.5 ± 0.9 Hz/sham *Intensity*: 0.7 ± 0.3 mA peak-to-peak *Duration*: 20 min *Electrode size*: target electrode 4 × 4 cm, return electrode 5 × 7 cm *Timing*: online	↗ (accuracy) Low risk
**Alpha and Beta**					
[59]	*N* = 19 (mean age: 24 ± 4 years) Crossover within-subject design	Temporal bisection task *Choice (long or short) Reaction times*	SMA (FC1) + FC2 *Device*: NeuroConn MagStim (NeuroCare, Ilmenau, Germany)	*Frequency*: 10 Hz/20 Hz/sham *Intensity*: 1.5 mA peak-to-peak *Duration*: 20 min *Electrode size*: both 5 × 5 cm *Timing*: online	↗ Low risk
**Multiple Frequencies**					
[60]	*N* = 36 (mean age: 24.4 ± 3.5 years) Crossover within-subject design	Flanker task *Speed-accuracy-trade off and reaction times*	Medial-frontal Cortex (FCz) + Pz *Device*: Neuroelectrics, Barcelona, Spain	*Frequency*: 2 Hz/6 Hz/11 Hz/21 Hz/60 Hz/sham *Intensity*: 1.5 mA peak-to-peak *Duration*: 20 min *Electrode size*: both 25 cm^2^ *Timing*: online	↗ (with theta-tACS) − (with other tACS) Low risk
**Decision-making**					
**Multiple Frequencies**					
[61]	*N* = 34 in total left frontal montage: *N* = 17 (mean age: 20.52 ± 2.52 years) right frontal montage: *N* = 17 (mean age: 21.17 ± 2.78 years) Between-subject design	Decision making task *Gains and losses and response times*	Left frontal cortex (F3) + ipsilateral shoulder Right frontal cortex (F4) + ipsilateral shoulder *Device*: BrainStim, EMS Medical, Bologna, Italy	*Frequency*: 5Hz/10 Hz/20 Hz/40 HZ/sham *Intensity*: 1 mA peak-to-peak *Duration*: 40 min *Electrode Size*: both 5 × 7 cm *Timing*: online	↘ (with beta-tACS to left hemisphere = riskier decisions) ↗ (with tACS to the right hemisphere) Low risk
**Procedural Memory (motor learning)**					
**Theta**					
[62]	*N* = 26 (mean age: 21.38 ± 1.52 years) Crossover within-subject design	Implicit motor learning task *Correct responses, reaction times*	Frontal area (FPz) + Pz *Device*: NeuroConn DC Stimulator Plus (NeuroCare, Ilmenau, Germany)	*Frequency*: 6 Hz/sham *Intensity*: 1 mA peak-to-peak *Duration*: 20 min *Electrode Size*: both 5 × 5 cm *Timing*: online	− Low risk
**Alpha**					
[63]	*N* = 20 (mean age: 23.8 ± 3.90 years) *N* = 15 (mean age: 61.66 ± 3.71 years) Between-subject design (older vs. younger participants)/Crossover within-subject design	Implicit motor learning task *Correct responses, reaction times*	Central area (C3) + supraorbital region *Device*: Eldith DC Stimulator, NeuroConn ((NeuroCare, Ilmenau, Germany)	*Frequency*: IAF/IAF + 2 Hz/sham *Intensity*: 1.5 mA peak-to-peak *Duration*: 10 min *Electrode Size*: target electrode 5 × 7 cm, return electrode 10 × 10 cm *Timing*: offline	− (accuracy) Reaction times: ↗ (in older adults with IAF and IAF + 2 Hz) ↗ (in younger adults with IAF + 2 Hz only) Low risk
**Alpha and Beta**					
[64]	*N* = 38 (mean age: 23 ± 3 years) Crossover within-subject design	Movement selection task *Duration between stimulus onset and verbal responses*	Central area (C3 or C4) + Pz *Device*: NeuroConn DC Stimulator Plus (NeuroCare, Ilmenau, Germany)	*Frequency*: 10 Hz/20 Hz/sham *Intensity*: 1.1 ± 0.3 mA peak-to-peak (range: 0.4–1.9 mA) *Duration*: ~30 min *Electrode Size*: both 5 × 5 cm *Timing*: online	↗ (with alpha-tACS ipsilateral to requested hand) − (with beta-tACS) Low risk
**Beta and Gamma**					
[65]	*N* = 20 (mean age: 24.9 years) Crossover within-subject design	Cued reaction time task *Reaction times*	Central area (left M1) + supraorbital region *Device*: NeuroConn DC Stimulator (NeuroCare, Ilmenau, Germany); BiStim (MagStim)	*Frequency*: IBF (or fixed 20.1 Hz)/75 Hz/sham *Intensity*: 0.69 ± 0.11 mA peak-to-peak; 1.3 ± 0.36 mA peak-to-peak *Duration*: 20 min *Electrode Size*: both 5 × 7 cm *Timing*: online	↗ (with gamma-tACS) − (with beta-tACS) Low risk
**Gamma**					
[66]	*N* = 17 (mean age: 24.5 ± 3.5 years) Crossover within-subject design	Implicit motor learning task *Reaction times*	Central area (left M1) + right supraorbital region *Device*: NeuroConn DC Stimulator (E.M.S., Bologna, Italy)	*Frequency*: 1 Hz/40 Hz/sham *Intensity*: 2 mA peak-to-peak *Duration*: 5.08 ± 1.13 min *Electrode Size*: both 5 × 5 cm *Timing*: online	↘ (with gamma tACS) − (with other tACS) Low risk
**Multiple Frequencies**					
[67]	*N* = 55 (mean age: 32.7 ± 6.8 years) Crossover within-subject design	Implicit motor learning task *Correct responses, Reaction times*	Central area (C3) + supraorbital *Device*: NeuroConn DC Stimulator Plus (NeuroCare, Ilmenau, Germany)	*Frequency*: 10 Hz/20 Hz/70 Hz/sham *Intensity*: 1 mA peak-to-peak *Duration*: 10 min *Electrode Size*: both 5 × 7 cm *Timing*: offline	↗ (with gamma-tACS) − (with other tACS) Low risk
**Working memory**					
**Theta**					
[68]	*N* = 25 (mean age: 23.5 ± 2.9 years) Crossover within-subject design	N-back task *Correct responses, reaction times*	Frontal/parietal cortex *Sync*: AF3/P3 in-phase + AF4/P4 in-phase *Desync*: AF3/AF4 in-phase + P3/P4 in-phase *Device*: Starstim (Neuroelectrics, Barcelona)	*Frequency*: 6 Hz/sham *Intensity*: 2 mA peak-to-peak *Duration*: 17–19 min *Electrode Size*: 3.14 cm^2^ (round-shaped) *Timing*: online	↘ (with anti-phase tACS) − (with in-phase tACS) Low risk
[69]	**Exp. 1** *N* = 18 (mean age: 29.8 ± 8.3 years) **Exp. 2** *N* = 14 (mean age: 21.9 ± 5.9 years) Crossover within-subject design	Delayed match-to-sample task *Working memory capacity*	Parietal cortex (P4) + (Cz/Oz/T8) *Device*: Starstim (Neuroelectrics, Barcelona)	*Frequency*: 4 Hz/7 Hz/sham *Intensity*: **Exp. 1** 1 mA peak-to-peak; **Exp. 2** 2 mA peak-to-peak *Duration*: at least 6 min per condition *Electrode Size*: target electrode 19.6 cm², each return electrode 4.9 cm² *Timing*: online	↗ (with low theta-tACS, for items presented contralateral to stimulation site only) − (with high theta-tACS) Low risk for **Exp. 1** High risk for **Exp. 2** (small sample size)
[70]	*N* = 36 (mean age: 20 years ± 4.25 months) Between-subject design	Corsi block-tapping task, Digit-span task, N-back task *Working memory capacity, correct responses (n-back only)*	Left parietal cortex (P3) + supraorbital region Frontal cortex (F3) + supraorbital region Right parietal cortex (P4) + supraorbital region *Device*: NeuroConn DC Stimulator (NeuroCare, Ilmenau, Germany)	*Frequency*: ITF/sham *Intensity*: 1–2.25 mA peak-to-peak *Duration*: 15 min *Electrode Size*: both 5 × 7 cm *Timing*: offline	↗ Low risk (but imbalanced gender distribution: 27 females)
[71]	*N* = 48 (mean age: 23 years) Crossover within-subject design	Visuo-spatial working memory task *Working memory capacity*	Parietal cortex (P3/P4) + left cheek **Exp. 1** in-phase between P3 and P4 **Exp. 2** anti-phase between P3 and P4 *Device*: NeuroConn DC Stimulator MC (NeuroCare, Ilmenau, Germany)	*Frequency*: 6 Hz/sham *Intensity*: 1.6 mA peak-to-peak *Duration*: 20–24 min *Electrode Size*: target electrode 4 × 4 cm, return electrode 5 × 7 cm *Timing*: online	↗ (with in-phase for low performer) − (with anti-phase or in-phase for high performer) Low risk
[72]	*N* = 46 (age range: 22–30 years) **Exp. 1**: *N* = 10 **Exp. 2**: *N* = 18 **Exp. 3** (control): *N* = 18 Crossover within-subject design	Letter discrimination task *Reaction times*	**Exp. 1**: no stimulation **Exp. 2/3**: Frontal/parietal cortex (F3/P3) + Vertex (Cz) *Device*: NeuroConn (NeuroCare, Ilmenau, Germany)	*Frequency*: **Exp. 1**: no stimulation; **Exp. 2**: 6 Hz in-phase or anti-phase/sham; **Exp. 3**: 35 Hz in-phase or anti-phase/sham *Intensity*: **Exp. 2/3** 1 mA peak-to-peak *Duration*: **Exp. 2/3** 14 ± 1.5 min *Electrode size*: **Exp. 2/3** all 5 × 5 cm *Timing*: **Exp. 2/3** online	↗ (with in-phase theta-tACS) − (with anti-phase theta- or gamma-tACS) Low risk
[73]	**Exp. 1** *N* = 10 (mean age: 28.6 ± 5.0 years) **Exp. 2** *N* = 21 (mean age: 27.38 ± 4.56 years) Crossover within-subject design	Working memory tasks **Exp. 1** 2-back/1-back vs. choice reaction task **Exp. 2** 2-back vs. choice reaction task + MRT *Correct responses, reaction times*	**Exp. 1** and **Exp. 2** frontal/parietal areas (F4/P4) + T8 **All Exp.** either in-phase or anti-phase between F4 and P4 *Device*: NeuroConn DC Stimulator MR (NeuroCare, Ilmenau, Germany)	*Frequency*: 6 Hz/sham *Intensity*: 1 mA peak-to-peak *Duration*: **Exp. 1**: 26.5 min; **Exp. 2**: 20 min in short runs of 20 s or 30 s *Electrode Size*: all 5 cm diameter (‘donut’-shaped) *Timing*: online	↗ (for choice reaction task with in-phase tACS) − (accuracy) High risk in **Exp. 1** (small sample size) Low risk in **Exp. 2**
[74]	*N* = 2 × 16 (mean age: 28.3 years (±7.6)/22.8 years (±5.2) Between-subject design (experimental vs. control montage)/Crossover within-subject design	Visuo-spatial working memory task *Correct responses, K-Value (Load x (Hits – False))*	*Target stimulation*: Parietal cortex (P4) + supraorbital region *Active control*: Parietal cortex (P4) + Vertex (Cz) *Device*: Magstim DC-Stimulator Plus	*Frequency*: 4 Hz/7 Hz/control (5.5 Hz) *Intensity*: 1.24 ± 0.3 mA peak-to-peak *Duration*: ~12 min *Electrode Size*: both 5 × 5 cm *Timing*: online	for arrays pointing to the left: ↗ (with 4 Hz-tACS) ↘ (with 7 Hz-tACS) − (for arrays pointing to the right) Low risk
**Alpha**					
[75]	*N* = 22 (mean age: 24.1 years) Between-subject design	Spatial updating paradigm *Updating bias*	Left parieto-occipital cortex (P3-O1) + vertex (Cz) Right parieto-occipital cortex (P4-O2) + vertex (Cz) *Device*: NeuroConn (NeuroCare, Ilmenau, Germany)	*Frequency*: 10 Hz *Intensity*: 1 mA peak-to-peak *Duration*: 25 min *Electrode size*: target electrode 4 × 3 cm, return electrode 9 × 5 cm *Timing*: online	↗ Low risk
**Gamma**					
[76]	*N* = 18 (mean age: 29.3 ± 7.65 years) Crossover within-subject design	N-back tasks *Correct responses, reaction times*	Frontal cortex (F3) + subraorbital region *Device*: NeuroConn DC Stimulator Plus (NeuroCare, Ilmenau, Germany)	*Frequency*: 40 Hz/tDCS/sham *Intensity*: 1.5 mA peak-to-peak *Duration*: 20 min *Electrode Size*: both 5 × 7 cm *Timing*: online	↗ (for 3-back condition when applied offline) − (with online-tACS, as well as for the 1- and 2-back condition) High risk (small sample size)
**Gamma and Theta**					
[77]	**Exp. 1** *N* = 20 (mean age: 21 years) **Exp. 2** *N* = 20 (mean age: 23 years) **Control 1** *N* = 20 (mean age: 23 years) **Control 2** *N* = 20 (mean age: 23 years) Crossover within-subject design	Visuo-spatial working memory task *d′*	Centro-to-temporal areas (CP1/T5) + right cheek **Exp. 1** anti-phase between CP1 and T5 **Exp. 2** in-phase between CP1 and T5 **Control 1** in-phase between CP1 and T5 **Control 2** anti-phase between CP1 and T5 *Device*: NeuroConn DC Stimulator MC (NeuroCare, Ilmenau, Germany)	*Frequency*: **Exp. 1/2**: 40 Hz/sham; **Control 1/2**: 6 Hz/sham *Intensity*: 1.5 mA peak-to-peak *Duration*: 20 min *Electrode Size*: both target electrodes 5 × 5 cm; return electrode 5 × 7 cm *Timing*: online	↗ (with anti-phase gamma-tACS) − (with in-phase gamma-tACS or theta-tACS) Low risk
**Multiple Frequencies**					
[78]	*N* = 25 (mean age: 69.1 ± 4.5 years) **Exp. 1** no stimulation **Exp. 2/3** Crossover within-subject design	Retro-cue working memory paradigm *Correct responses*	Parietal cortex (P3) + P4	*Frequency*: **Exp. 1**: no stimulation; **Exp. 2**: 4 Hz/10 Hz/40 Hz; **Exp. 3**: 10 Hz/sham *Intensity*: 1.5 mA peak-to-peak *Duration*: 20 min *Electrode Size*: both 5 × 7 cm *Timing*: online	↗ (with alpha-tACS but only in **Exp. 2**, no replication in **Exp. 3**) − (with other tACS) Low risk
**Declarative memory**					
**Theta**					
[79]	*N* = 12 young (YA, mean age: 22.3 ± 1.5 years) *N* = 12 older adults (OA, mean age: 66.3 ± 3.9 years) Crossover within-subjects design	Language-learning paradigm *Correct responses*	Centro-parietal area (CP5) + supraorbital region *Device*: NeuroConn (NeuroCare, Ilmenau, Germany)	*Frequency*: 6 Hz/sham *Intensity*: 1 mA peak-to-peak *Duration*: 20 min *Electrode size*: target electrode 5 × 7 cm, return electrode 10 × 10 cm *Timing*: online	↗ (in older adults) − (in younger adults) High risk (small sample size)
[80]	*N* = 72, (*N* = 24 for each experiment) **Exp. 1** (mean age: 23.5 ± 3.1 years) **Exp. 2** (mean age: 24.3 ± 2.9 years) **Exp. 3** (mean age: 23.2 ± 2.2 years) Between-subject design	Paired-associative learning task *Correct responses*	Left temporal cortex (T7) + (FPz/T8) *Device*: NeuroConn (NeuroCare, Ilmenau, Germany)	*Frequency*: 5 Hz modulated bursts of 80 Hz gamma/sham *Intensity*: 1 mA peak-to-baseline *Duration*: 10 min *Electrode size*: all 3 cm² *Timing*: online	↘ (with gamma-bursts coupled with troughs of theta-waves) − (with every other coupling) *Low*
[81]	*N* = 28 (mean age: 71.2 ± 6.4 years) Crossover within-subject design	Face-occupation task (Paired-associative learning) *Correct responses*	Ventrolateral PFC (intersection of T3-F3 and F7-C3 and midpoint F7-F3) + supraorbital region *Device*: NeuroConn (NeuroCare, Ilmenau, Germany)	*Frequency*: 5 Hz/sham *Intensity*: 1 mA peak-to-peak *Duration*: 20 min *Electrode size*: target electrode 5 × 7 cm, return electrode 10 × 10 cm *Timing*: online	− Low risk
[82]	*N* = 25 (age range: 18–28 years) Crossover within-subjects design	Short-term recognition task *Correct responses* Long-term memory recognition task *Strength of recollection and familiarity*	*Target stimulation*: Right parietal cortex (P4) + (T8/C2/CP1/Oz) *Active control*: Left parietal cortex (P3) + (T7/C1/CP2/Oz) *Device*: Multi-channel stimulator (StarStim, Neuroelectrics, Barcelona, Spain)	*Frequency*: 4 Hz/sham *Intensity*: 3 mA peak-to-peak *Duration*: 20 min *Electrode size*: 5 round-shaped electrodes with 1 cm radius *Timing*: online	↗ (with theta-tACS; for familiarity) Low risk
[83]	*N* = 60 (age range: 18–45 years) *N* = 19 (mean age: 28.4 ± 6.9 years) *N* = 21 (mean age: 25.3 ± 5.6 years) *N* = 19 (mean age: 24.9 + 4.7 years) Between-subject design	Visual associative memory task *Correct responses*	Right fusiform cortex (P10) + (FP1/P2/P3/PO7) *Device*: Soterix MxN TES device (Soterix Medical Inc., New York, USA)	*Frequency*: 6 Hz/sham *Intensity*: 2 mA peak-to-baseline *Duration*: 10 min *Electrode size*: HD-tES electrodes *Timing*: online	↗ (accuracy and forgetting) Low risk
**Gamma**					
[84]	*N* = 70 (mean age: 22.12 + 2.16 years) *Congruent group 60-60* (*N* = 17) *Congruent group 90-90* (*N* = 18) *Incongruent group 60-90* (*N* = 17) *Incongruent group 90-60* (*N* = 18) Between-subject design/Crossover within-subjects design	Word-list learning *Correct responses;* *d’; Reaction times*	Dorsolateral prefrontal cortex (F3) + left wrist *Device*: NeuroConn DC Brain Stimulator Plus (NeuroCare, Ilmenau, Germany)	*Frequency*: 60 Hz/90 Hz/sham *Intensity*: 1.5 mA peak-to-peak *Duration*: 15 min during encoding, 15 min during retrieval *Electrode size*: both 5 × 7 cm *Timing*: online	↗ (with the same gamma-tACS during encoding and retrieval) − (with different gamma-tACS during encoding and retrieval) Low risk
[85]	*N* = 36 (mean age: 21.3 ± 0.5 years) *N* = 18 (mean age: 21.2 ± 0.4 years) *N* = 18 (mean age: 21.3 ± 0.5 years) Between-subject design, mixed with repeated-measures within-subject design	Episodic memory task (Word learning and recognition) *Correct responses,* *d’, Reaction times*	PFC (F3) + left wrist *Device*: Eldith Brain Stimulator (NeuroCare, Ilmenau, Germany)	*Frequency*: 60 Hz/sham *Intensity*: 0.75 mA peak-to-peak *Duration*: 15 min during encoding, 15 min during retrieval *Electrode size*: both 5 × 7 cm *Timing*: online	↗ (accuracy) − (reaction times) Low risk
[86]	**Exp. 1** *N* = 36 (mean age: 20.03 ± 2.38) **Exp. 2** *N* = 36 (mean age: 20.97 ± 2.22) Crossover within-subject design	Episodic memory task (Word and face learning and recognition) *Correct responses*	Left IFG (FP1) + C5 Right IFG (FP2) + C6 *Device*: 4-channel DC Stimulator MC (NeuroCare, Ilmenau, Germany)	*Frequency*: 18.5 Hz/control frequencies: 6.8 Hz, 10.7 Hz, 30 Hz, 48 Hz/sham *Intensity*: **Exp. 1**: 2 mA peak-to-peak **Exp. 2**: 1.6 mA peak-to-peak *Duration*: event-related stimulation for 2 s during each stimulus presentation (2 s × 360 = 720 s total stimulation duration) *Electrode size*: **Exp. 1**: 4 donut-shaped electrodes, each 14 cm^2^; **Exp. 2**: 4 round electrodes, each 10.75 cm^2^ *Timing*: online	− Low risk

Note. tACS, transcranial alternating current stimulation (HD-tACS, high-definition tACS); tDCS, transcranial direct current stimulation (HD-tDCS, high-definition tDCS); SMA, supplementary motor area; Exp., experiment; Hz, Hertz; IAF, individual alpha frequency; ITF, individual theta frequency; IBF, individual beta frequency; mA, Milliampere; min, minutes. Low risk bias = Sufficient sample size and control conditions (sham condition and/or control frequencies), description of blinding procedure and efficiency, randomization procedure described, detailed stimulation protocol and description of statistical analysis. High risk bias = Small sample size, no control parameters. Electrode positions were reported as follows: target electrode(s) + return electrode(s). The direction of the arrows indicates improvement (↗), decrease (↘), or no change (−) in performance with tACS.

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
