# Peer review of "The Modulation of Cognitive Performance with Transcranial Alternating Current Stimulation: A Systematic Review of Frequency-Specific Effects"

_brainsci, 2020, doi:10.3390/brainsci10120932_

Round 1
Reviewer 1 Report
In this review, Klink and colleagues examine frequency-specific tACS effects on cognitive function in healthy adults. Through compilation and analysis of 57 studies, they determine that working memory, executive function, and declarative memory benefitted from theta tACS, whereas auditory and visual perception were improved with gamma tACS. Attentional results were less clear; however, the authors suggest that there is an improvement with alpha or gamma tACS. Overall, the study is well written, comprehensively addresses an important question, and compiles results in a useful way in an ever-growing field of research. I only have one minor comment (below) for the authors to address.
Methods
In the inclusion criteria, please provide rationale for your choice of n=15 for within-subject designs and n=20 for between-subject designs.
Reviewer 2 Report
This is a very well written and interesting systematic review.
However, I have some comments/suggestion which may improve the quality of this paper:
Can the authors please comment on the following points:
- Long-term effects of tACS
- Sex differences of tACS
- This reviewer thinks, that most of the tACS and also tDCS improvements are placebo effects. Most of the current (~75% is not reaching the brain).
- Was blinding effective in the studies?
Author Response
Please see the attachement.

Reviewer 3 Report
It was a great pleasure to read the review article by Klink et al. regarding the various cognitive effects of transcranial alternating current stimulation (tACS). Although the review is quite extensive, I feel that it is missing some information that would make it more useful for the readers.
1) In table 1, there should be a column indicating the neurocognitive effects in simple terms. For example, an arrow upward if cognition improved, downward if there was a decrease, and a horizontal line if no effects.
2) Could the authors make some suggestions: what would be the minimum recommended tACS duration and intensity? Or any other recommendations based on the studies they have reviewed?
3) State-dependency is a hot topic in all brain stimulation studies. As the authors reviewed both studies that were conducted “online” and “offline,” were there any differences in the outcomes?
Author Response
Please see the attachement.
